# Encapsulation of Thymol and Eugenol Essential Oils Using Unmodified Cellulose: Preparation and Characterization

**DOI:** 10.3390/polym15010095

**Published:** 2022-12-26

**Authors:** Koranit Shlosman, Dmitry M. Rein, Rotem Shemesh, Na’ama Koifman, Ayelet Caspi, Yachin Cohen

**Affiliations:** 1The Interdepartmental Program in Polymer Engineering, Technion-Israel Institute of Technology, Haifa 32000, Israel; 2R&D and Customer Service Department Carmel Olefins Ltd., Haifa 31014, Israel; 3Department of Chemical Engineering, Technion-Israel Institute of Technology, Haifa 32000, Israel; 4Tama Group, Kibbutz Mishmar Ha’emek 1923600, Israel

**Keywords:** cellulose, essential oils, encapsulation, environmentally friendly pesticides

## Abstract

Essential oils (EOs) are volatile natural organic compounds, which possess pesticidal properties. However, they are vulnerable to heat and light, limiting their range of applications. Encapsulation of EOs is a useful approach to overcome some of these limitations. In this study, a novel emulsification technique is utilized for encapsulation of thymol (TY) and eugenol (EU) (EOs) within microcapsules with an unmodified cellulose shell. Use of low cost materials and processes can be beneficial in agricultural applications. In the encapsulation process, unmodified cellulose was dissolved in 7% aqueous NaOH at low temperature, regenerated to form a dispersion of cellulose hydrogels, which was rigorously mixed with the EOs by mechanical mixing followed by high-pressure homogenization (HPH). Cellulose:EO ratios of 1:1 and 1:8 utilizing homogenization pressures of 5000, 10,000 and 20,000 psi applied in a microfluidizer were studied. Light microscopy, high-resolution cryogenic scanning electron microscopy (cryo-SEM) and Fourier transform infrared spectroscopy (FTIR) revealed successful fabrication of EO-loaded capsules in size range of 1 to ~8 µm. Stability analyses showed highly stabilized oil in water (O/W) emulsions with instability index close to 0. The emulsions exhibited anti-mold activity in post-harvest alfalfa plants, with potency affected by the cellulose:EO ratio as well as the EO type; TY showed the highest anti-mold activity. Taken together, this study showed potential for anti-fungal activity of cellulose-encapsulated EOs in post-harvest hay.

## 1. Introduction

Essential oils (EOs) are complex mixtures of volatile compounds formed by aromatic plants as secondary metabolites and characterized by strong odors. EOs are mainly composed of terpenoids, phenolic and aromatic compounds and their composition can vary, depending on the characteristics of the plant, the parts from which they are extracted, extraction method and environmental conditions [1,2]. They possess various biological properties, such as antioxidative, and are considered additives suitable for the food industry. Many EOs exhibit antimicrobial, insecticidal and herbicidal activity, and have the potential to inhibit or eliminate the presence of pathogenic microorganisms [3,4,5]. Antimicrobial activity against *Escherichia coli*, *Staphylococcus aureus* and *Bacillus subtilis* was reported for cinnamon, thyme, and clove EOs [6]. Antifungal effects against *Penicillium parasiticus* and *Aspergillus niger* and insecticidal activity, have also been reported [6]. EOs were also shown to inhibit *Butrytis cinerea* growth, the cause for gray mold in postharvest strawberries [5]. Thanks to their broad antimicrobial activity, EOs can be used in a variety of applications, including food, cosmetics, and pharmaceutics [6,7,8]. It is assumed that the antimicrobial activity and strength of EOs are a function of their chemical structure and their chemical interactions with pathogens [9]. Thymol (TY), found in leaves of oregano and thyme, and eugenol (EU) found in basil, thyme, clove, and pimento leaves, are EOs that have been extensively investigated for their antibacterial, antioxidative and anti-inflammatory activity [10,11,12]. TY and EU differ in their chemistry: while TY is an oxygenated terpene, EU is a phenylpropanoid [13]. The chemical structures of TY and EU are depicted in Figure 1a,b, respectively. 

TY and EU are widely used as flavoring agents in food as well as in the pharmaceutical, cosmetic and perfume industries. Yet, their volatility renders them vulnerable to oxidation, light and heat, and affect their composition, therefore limiting their range of applications [3]. Moreover, in various applications, such as medical, agricultural and cosmetic, controlled release is required to avoid toxicity and to provide for long-term antimicrobial activity [14]. EOs can also be incorporated into plastic articles such as containers, films and bags to be used in food packaging or agricultural applications. Use of such active packaging may increase food safety, and reduce food loss and the use of food preservatives. It can also decrease mold- and bacteria-related crop damage [1]. Encapsulation is an attractive technology for protecting sensitive ingredients (gas, liquids or solids) such as flavorants, vitamins and aromas, and for minimizing undesirable oxidation. Encapsulation also provides control over the release of volatile compounds thereby extending their possible uses [15]. 

Emulsification is a process in which two immiscible liquids, typically oil and water, form a (semi)stable dispersion, such as oil droplets in water (O/W emulsions) [6]. Stabilization of disperse droplets in the continuous phase can be achieved using an emulsifier, i.e., amphiphilic surface-active molecules, polymers or solid particles which adsorb to the interface between the two liquids [16]. Increased consumer awareness and environmental commitment, together with regulations relating to currently used synthetic emulsifiers (e.g., sorbitan esters) and those of animal origin (e.g., gelatin), have spurred an increased interest in natural and plant-based emulsifiers. Biopolymers (proteins and polysaccharides, for example), biosurfactants (e.g., glycosides and glycolipids) and bioparticles are examples of natural emulsifiers being extensively investigated. Bioparticles are derived from biopolymers, such as chitosan, starch, lignin, cellulose or proteins [17]. In contrast to surfactants, which are dynamically adsorbed and desorbed at the interface, bioparticles adsorb irreversibly to it, forming the so-called “Pickering emulsions” [18,19]. While stabilization by the Pickering mechanism is widely used [17,18,20], it is less suited for encapsulation of EOs in applications that require a more continuous protective coating [21,22,23].

Cellulose is a linear polysaccharide, a homopolymer of glucose, found in all plants and characterized by a semicrystalline fibrillar structure comprised of highly ordered crystalline domains separated by disordered amorphous regions [24]. Its intrinsic amphiphilic character, as well as its mechanical strength, motivated evaluation of its application as an emulsifier or encapsulating material which can form a coating shell at oil–water interfaces [25,26]. Under appropriate conditions, “Pickering emulsions” can be formed by assembly of nano- or micro-crystalline cellulosic solids at the oil-water interface [16,27,28]. Alternatively, cellulose can form a continuous amorphous coating when applied from solution by high shear mixing with oil and water [25,29]. In such case, encapsulation has been suggested to occur upon cellulose regeneration from solution at the interface between the cellulose solution and the aqueous medium [29]. A much simpler solution-regeneration process was developed, which does not require costly solvents such as ionic liquids. In this process, cellulose is dissolved in aqueous NaOH (~7 wt.%) at temperatures below 0 °C, and regenerated in water to form a dispersion of hydrogel particles. High-shear mixing of the hydrogel suspension with hydrophobic liquids results in their encapsulation by a continuous cellulose coat and in formation of stable emulsions [30,31,32]. Waste material from textiles or the paper industry was also shown to be a suitable raw material for this process [30]. Several works have studied encapsulation of EOs by modified cellulose (e.g., methyl-cellulose, ethyl-cellulose, carboxy-methyl-cellulose or hydroxypropyl-cellulose) [25,33]. However, it is of interest to apply unmodified cellulose for EOs encapsulation, to eliminate the need for chemical modification. Previous studies utilized the regenerated cellulose hydrogel method to fabricate cellulose coated micro-capsules with thick coatings for its subsequent hydrolysis. The present study aimed to evaluate the encapsulation of TY and EU EOs by unmodified cellulose, applied as the shell material, using the hydrogel encapsulating technique, to form stable aqueous emulsions. In addition, the feasibility of maintaining the antifungal activity of the EOs was assessed.

## 2. Materials and Methods

### 2.1. Materials

Microcrystalline cellulose (MCC) powder (Avicel^®^ batch No. MKCJ3230) with particle sizes in the range of 70–250 µm, was purchased from Sigma Aldrich Co., Rehovot, Israel. NaOH, thymol (98.5%) and eugenol (98%) were purchased from Sigma Aldrich. Deionized water was used in this work.

### 2.2. Methods

#### 2.2.1. Fabrication of Encapsulated Essential Oil Emulsions

A cellulose hydrogel microparticle dispersion was prepared as follows: 4.2 wt.% MCC was dissolved in 7 wt.% NaOH in deionized water. After 5 min of mixing with a magnetic stirrer, the solution was introduced into a −17 °C cooling bath (LT ecocool 100, Grant Instruments, Shepreth Cambridgeshire, UK) under an overhead stirrer set at 500 rpm, for about 10 min (HsiangTai Machinery Industry, New Taipei City, Taiwan) until the hazy solution turned clear. Then, the solution was poured into deionized water to form a hydrogel microparticle dispersion, which was then washed with deionized water until very low conductivity (<0.7 mS) was measured. Hydrogel content was determined gravimetrically, in triplicates.

Emulsions were prepared as follows: Hydrogel dispersion and deionized water were mixed to obtain a cellulose content of 1.5 wt.%. The EO (TY or EU) was then added to the mixture in the ratios detailed in Table 1. The EO/hydrogel/water mixtures were mixed in a mechanical homogenizer (T18 digital Ultra-Turrax, IKA Works, Staufen, Germany) for 10 min at 15,000 rpm and then subjected to high-pressure homogenization (Model LM-20 microfluidizer, Microfluidics, Newton, MA, USA), under 5000, 10,000 and 20,000 psi for 10 min. High shear and pressure applied during homogenization processes ensure intimate contact between the cellulose and EOs, resulting in a continuous coating surrounding the EO. When TY (which is solid at room temperature) was used, it was first melted on a hot plate (~50 °C for 5 min) before it was added to the hydrogel mixture and continuously heated (~50 °C) during the mechanical homogenization. Moreover, sufficient temperature is enabled during the high pressure homogenization (HPH), due to the high shear process. 

#### 2.2.2. Characterization

##### Scanning Electron Microscopy 

Microcapsule morphology was studied by cryogenic-temperature, high-resolution scanning electron microscopy (cryo-HR-SEM) using a Zeiss Ultra Plus (Carl Zeiss, Jena, Germany) high-resolution scanning electron microscope equipped with a Schottky field-emission gun and a BalTec VCT100 (Balzers, Liechtenstein) cold-stage, which was maintained below −145 °C. Specimens were imaged at a low acceleration voltage of 1.0–1.1 kV and working distances of 3.7–6.4 mm. The Everhart Thornley (“SE2”) and the In-the-column (InLens) secondary electron imaging detectors were used for morphological characterization purposes. The energy-selective backscattered (“ESB”) electron detector was used for compositional characterization. Images were acquired by the SmartSEM software and analyzed by ImageJ (Carl Zeiss, Jena, Germany).

Specimens were prepared by the drop plunging method, where a 3 μL drop of solution is set on top of a special planchette which maintains the droplet shape and is manually plunged into liquid ethane at its freezing point (−183 °C), after which it is set on top of a specialized sample table. The frozen droplets are transferred to a BAF060 (Leica Microsystems, Wetzlar, Germany) freeze fracture system, where they were fractured by a rapid stroke of a cooled knife, exposing the inner part of the drop. They were then transferred to the pre-cooled HR-SEM and imaged as described above. Ideally, imaging was performed as close as possible to the drop surface, where the cooling rate is expected to be maximal.

##### Light Microscopy 

Samples were observed using a light microscope equipped with a bright-field collector lens unit using a transmission light source (Olympus BX60, Evident Corp., Waltham, MA, USA). A small sample of each emulsion was placed on a glass slide and covered with a cover-slip. Images were recorded and analyzed with Stream Essentials software (Olympus Scientic Solutions, Evident Corp., Waltham, MA, USA). 

##### Attenuated Total Reflectance Fourier Transform Infrared Spectroscopy (ATR-FTIR)

ATR-FTIR spectroscopy was used to determine the presence of functional groups of neat EOs and MCC in the emulsions. The infrared spectra of the samples were recorded using a Perkin Elmer Fourier transform infrared spectrophotometer (Perkin Elmer FTIR Spectrum BX II, Waltham, MA, USA) in attenuated total reflectance (ATR) mode, in a spectral range of 5000–500 cm^−1^ at a resolution of 4 cm^−1^. Analysis was performed using the Perkin Elmer spectrum IR version 10.6.1 software. To avoid interference, all samples underwent centrifugation, in order to dramatically decrease their water content.

##### Emulsion Stability and Particle Size Distribution

The physical stability of emulsions was evaluated by an analytical centrifugal analyzer (LUMiSizer, L.U.M. GmbH, Berlin, Germany) which measures time- and space-resolved transmission profiles throughout the analyzed sample. The parameters used for the measurement were temperature: 25 °C, time interval: 30 s, experiment time: 24 h, rotational speed: 2000 rpm. Absorbance of 865 nm light was recorded. The particle size distribution was also measured by LUMiSizer operated at the same parameters.

##### Emulsion Anti-Mold Activity

The anti-mold activity of the fabricated emulsions was tested by observing mold development in a model system of post-harvest alfalfa plant. Alfalfa plant (Medicago sativa; leaves and stems, 3 g) was placed in a 90 mm Petri dish and wetted with 4 mL water. Emulsions (5 mL) were then smeared on a paper sheet which was placed at the inner side of a Petri dish cover. The systems (including a control- with no emulsion placed) were incubated in a climate chamber at 27 °C and 60% humidity in order to induce in vitro mold and pathogenic rapid growth. Observations for mold growth were made daily and included weighing of the Petri dish so as to replenish the evaporated water. Mold size was measured quantitatively using a ruler and percentage of mold growth was calculated.

## 3. Results and Discussion

Microcapsules of EOs encapsulated by a cellulose coating were fabricated at two cellulose:EO weight ratios, using HPH at three different pressures, as detailed in Table 1. Previous studies have shown that the main effects of the cellulose:oil ratio are controlling the size and shell thickness of the cellulose encapsulated particles. In these studies, emulsions were fabricated by the same procedure used in the current research [30,31,32]. A thick coating was desired in the previous studies, since enzymatic cellulose hydrolysis was considered. For this aim, a 1:1 cellulose:oil ratio was found to be most suitable. In contrast, the current study sought to generate a thinner coating for the purpose of EO encapsulation and subsequent release. A higher EO loading, i.e., a higher cellulose:EO ratio was thus required, while avoiding excessive EO content, which may cause a significant amount of unencapsulated EO. Therefore, emulsions fabricated with 1:1 and 1:8 cellulose:EO ratios were chosen. 

### 3.1. Morphological Observations by Scanning-Electron and Light Microscopies

Electron microscopy analysis of the microcapsule morphology, e.g., shape, dimensions and wall thickness is presented in Figure 2 and Figure 3. Cellulose-coated capsules loaded with either EU or TY were spherical, well dispersed throughout the sample and exhibited a morphology that was independent of the HPH pressure employed. At a ratio of 1:8 cellulose:EO (Figure 2), microcapsules were characterized by a diameter of about 4–6 µm and a well-defined wall approximately 50–60 nm thick. Dispersed cellulose hydrogel particles were also observed in different areas of all samples. Interestingly, in some cases the EO core was detached from the cellulose shell during cryo-fracturing, which provided a view of their inner shell side (Figure 2A,F). A cellulose network, originating from the hydrogel was clearly observed inside the capsule. In the background, small crystals of water redeposited on the sample surface during the water sublimation process after cryofracture showed up as white dots. Similar structures, albeit with a thicker inner shell, were reported in previous studies [29,31,32].

When cellulose:EO ratio was 1:1 (Figure 3), cellulose-coated microcapsules containing TY or EU fabricated by HPH at 10,000 psi were spherical and characterized by a diameter of about 1 µm, significantly smaller than those fabricated at the 1:8 ratio.. These microcapsules appeared embedded between cellulose tangles, as there is a high amount of free cellulose, originating from the hydrogel, evident in the background of all the images. 

In some cases, non-spheroidal structures were observed, as shown in Figure 4 for the EU-1:8-5k and EU-1:8-10k emulsions, which took on an ellipsoidal egg-like shape, in addition to spherical microcapsules with an apparent rigid shell. An external hydrogel layer, several hundred nanometers thick, seemed to surround the encapsulated particles. It seems that cellulose hydrogel adsorbed to an existing capsule form a soft corona around it, which is deformed during the high shear processing.

Light microscopy was used to complement the morphological characterization. Images of emulsions showed that the pressure applied in the HPH (at 5000, 10,000 and 20,000 psi) did not affect particle diameter (Figure 5). Furthermore, the microcapsules were shown to be well dispersed in the emulsion.

### 3.2. Attenuated Total Reflectance (ATR) Fourier Transform Infrared (FTIR) Spectroscopy

FTIR spectra in the range of 4000–500 cm^−1^ are presented in Figure 6A,B. MCC was characterized by absorption peaks in the range 3000–3700 cm^−1^ and at 902 cm^−1^, attributed to the presence of OH groups; at 2891 cm^−1^ and 1024 cm^−1^, attributed to C–H stretching vibration and at 1636 cm^−1^, attributed to the vibration of O-H bending of adsorbed water molecules [34,35]. TY FTIR spectra were characterized by absorption peaks at 3000–3500 cm^−1^, attributed to the presence of phenolic O-H; at 2865, 2926, 2958 and 2968 cm^−1^, attributed to a C-H bond and at 1418 and 1459 cm^−1^, attributed to C–H stretching vibrations [36]. The presence of an aromatic ring in TY was exhibited by its C-C stretching at 1622 cm^−1^ [33]. EU FTIR spectra were characterized by peaks at 1511, 1613 and 1638 cm^−1^, ascribed to stretching vibration originating from the aromatic C-C bond, and several peaks in the range of 1100–1300 cm^−1^ arising from the asymmetric stretch of C-O-C originating from the ether functional group [37,38]. Absorption bands characteristic of TY were clearly seen in the two emulsions containing this EO (Figure 6A). The additional transmission band was attributed to water (in the range of 3000–3700 cm^−1^). It was noted that the TY-characteristic absorption peaks obscured the peaks characteristic of MCC. Similarly, bands characteristic of EU were clearly seen in the two emulsions containing this EO (Figure 6B). Taken together, the ATR- FTIR analyses indicated successful incorporation of the EOs into cellulose microcapsules without chemical changes, as no new peaks appeared, and no characteristic peaks disappeared [7,39].

### 3.3. Emulsion Stability and Particle Size Distribution

Stability and particle size distribution of four emulsions (TY-1:8-10k, TY-1:1-10k EU-1:8-10k and EU-1:1-10k) were measured in duplicates by a LUMiSizer instrument. The emulsions were force-separated by subjecting them to accelerated centrifugal force while light absorbance through the sample as a function of time and position was recorded. The instability index was calculated as the differences in light transmission at predefined times and locations [40]. Instability index values range from 0 to 1, where low values, closer to 0, indicate a very stable emulsion, while high values, closer to 1, indicate an unstable emulsion. Instability values calculated for the above- mentioned emulsions indicated very high emulsion stability: 0.12 and 0.11 for TY-1:1-10k and EU-1:1-10k, respectively and 0.23 and 0.32 for TY-1:8-10k and EU-1:8-10k, respectively. Particle size distribution analyses supported the SEM findings, where cellulose:EO ratio 1:1 yielded much smaller capsules than the 1:8 ratio (2.3 and 1.8 µm for TY-1:1-10k and EU-1:1-10k, respectively and 9.3 and 4.9 µm for TY-1:8-10k and EU-1:8-10k, respectively). 

### 3.4. Emulsion Anti-Mold Activity

Mold development was investigated in samples of alfalfa plant exposed to different emulsions. Observations in the different systems are shown in Figure 7. While the control system showed 55% mold development in the first 3 days of incubation, all of the samples treated with emulsions showed reduced mold development. TY 1:8-10k exhibited excellent anti-mold activity, with no mold development throughout the entire 13 days of trial. EU 1:8-10k and TY 1:1-10k also showed very good anti-mold activity, with low mold development (9% and 36%, respectively) after 7 days of the trial. EU 1:1-10k exhibited poor anti-mold activity, with 36% mold development measured on day 3 of the trial. The differences in the anti-mold activity can be explained by the amount of EO in the emulsion, which is an outcome of the cellulose:EO ratio and by the difference in EO chemistry. Emulsions with a 1:8 ratio contain a higher amount of EO, and therefore release more of it to the head-space of the Petri dish, leading to improved anti-mold activity. Additionally, it can be seen that for the post-harvest alfalfa plant, a model system for hay, thymol had improved anti-mold activity compared to eugenol. This can be explained by the different chemistries of TY and EU. TY is an oxygenated terpene, while EU is a phenylpropanoid, leading to different biological interactions with mold-forming pathogens. Similar results were reported by Dewitte et al. [41], who studied the effect of different EOs on gray mold development in Lupin (*Lupinus* L.); they noted a major difference between the antifungal activity of the different EOs. 

## 4. Conclusions

TY and EU microcapsules with an unmodified cellulose coating were successfully fabricated using a novel method for producing O/W emulsions, precluding the need for additional substances such as surfactants. Two cellulose:EO ratios were investigated: 1:8 and 1:1. The former leading to much higher diameter (4–6 µm) than the latter (~1 µm). The pressure applied in the homogenization process did not have a significant effect on the particle size. The emulsions proved stable, at both cellulose:EO ratios. The emulsions were shown to inhibit mold growth in post-harvest alfalfa plant, at a potency that was dependent on the cellulose:EO ratio and the EO type. TY-based emulsions showed better anti-mold activity than the EU-based ones, with microcapsules prepared from 1:8 cellulose:TY emulsions exhibiting the best anti-mold activity. These results carry possible applicative potential as eco-friendly crop-protecting agents, in particular as anti-fungal agents in post-harvest hay.

## Figures and Tables

**Figure 1 polymers-15-00095-f001:**
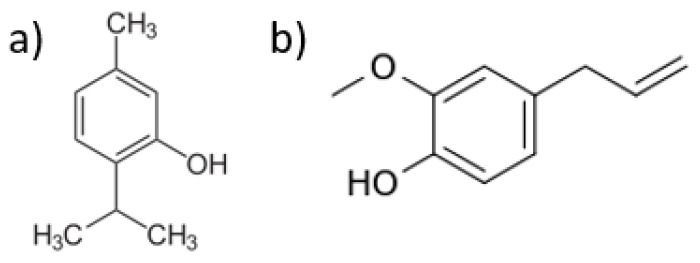
Chemical structure of (**a**) thymol and (**b**) eugenol.

**Figure 2 polymers-15-00095-f002:**
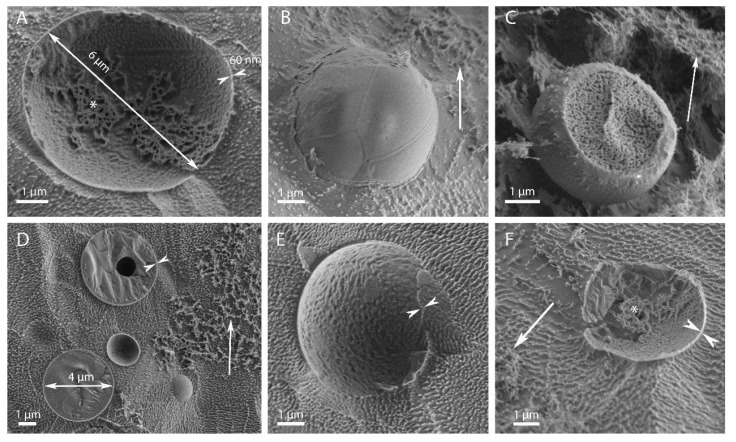
Cellulose microcapsules containing TY (**A**–**C**) or EU (**D**–**F**) at a 1:8 ratio, fabricated by HPH under (**A**,**D**) 5000 psi, (**B**,**E**) 10,000 psi, or (**C**,**F**) 20,000 psi. The outward and inward pointing double arrows indicate the capsule diameter and shell thickness, respectively. The single arrow points to the cellulose hydrogel particles in the background and the asterisks mark the cellulose network at the inner shell side after the EO core was detached during specimen fracture.

**Figure 3 polymers-15-00095-f003:**
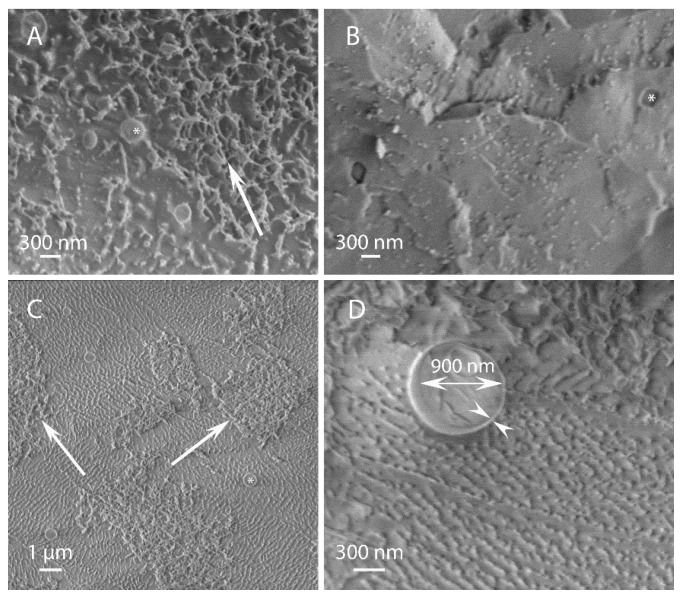
Cellulose coated microcapsule emulsions of TY (**A**,**B**) or EU (**C**,**D**) fabricated at a 1:1 cellulose:EO ratio, by HPH at 10,000 psi. The asterisks (in **A**–**C**) mark the microcapsules. The single arrows (in (**A**,**C**)) point to the cellulose network in the background. The outward and inward pointing double arrows (in (**D**)) indicate the capsule diameter and shell thickness, respectively.

**Figure 4 polymers-15-00095-f004:**
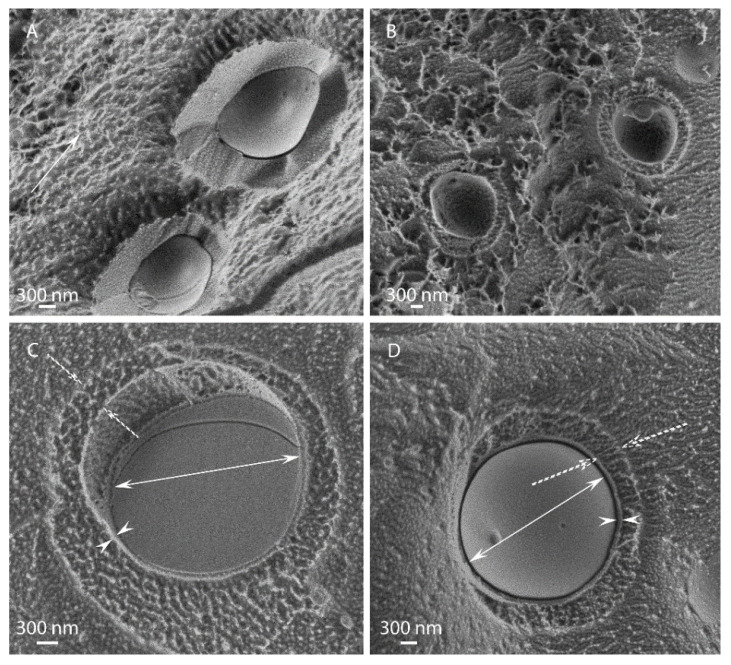
Cellulose capsules containing EU exhibiting egg-like structure at different magnifications fabricated by HPH under 5000 psi at cellulose:EO ratio 1:8. The single arrow (in (**A**)) points to cellulose network in the background, the outer andinner pointing double arrows indicate the capsule inner diameter and inner shell thickness, respectively, and the dashed arrows point the outer shell thickness.

**Figure 5 polymers-15-00095-f005:**
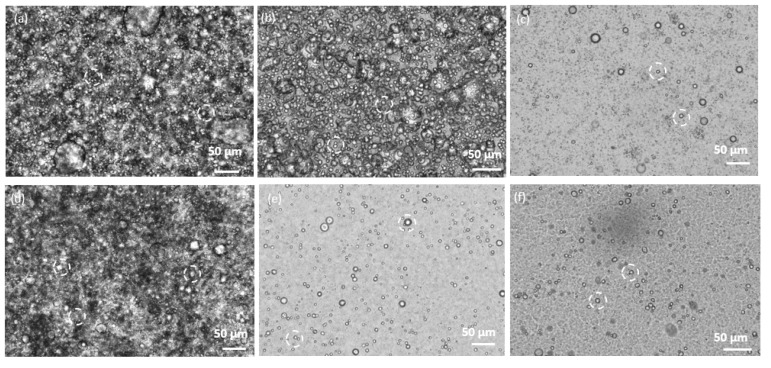
Light microscope images of cellulose- coated capsules containing TY (**a**–**c**) or EU (**d**–**f**) (cellulose:EO ratio 1:8), fabricated by HPH under (**a**,**d**) 5000 psi, (**b**,**e**) 10,000 psi (**c**,**f**) 20,000 psi. Particles are marked by dashed white circles.

**Figure 6 polymers-15-00095-f006:**
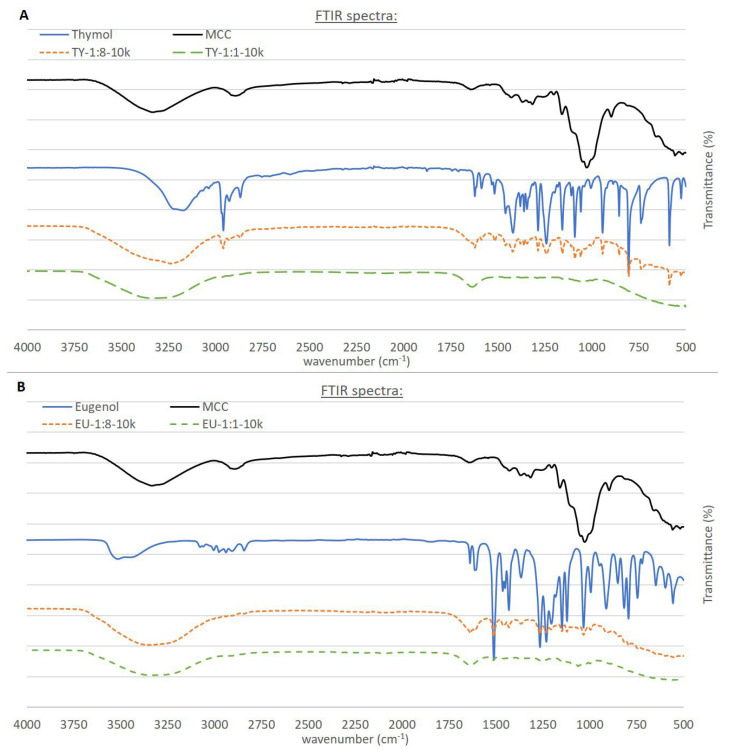
FTIR spectra of MCC and (**A**) neat TY, TY-1:8-10k and TY-1:1-10k and (**B**) neat EU, EU-1:8-10k and EU-1:1-10k.

**Figure 7 polymers-15-00095-f007:**
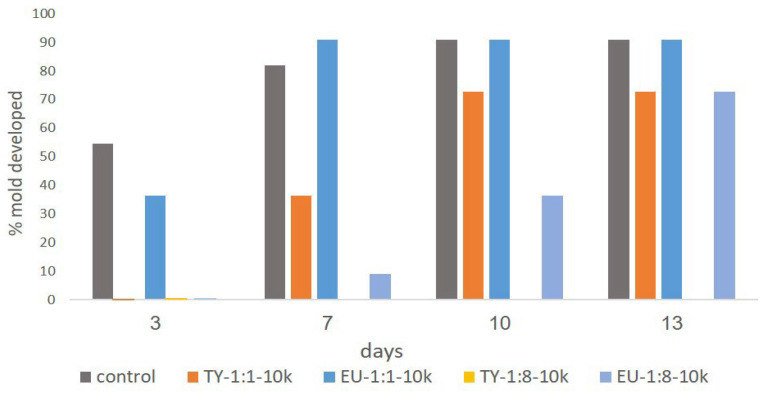
Mold development observations of alfalfa plant in a Petri dish exposed to the different emulsions. Observations performed after 3, 7, 10 and 13 days of incubation. A Petri dish with no emulsion was used as the control.

**Table 1 polymers-15-00095-t001:** Emulsions prepared.

Sample ID	EO ^1^ Type	Cellulose: EO Ratio wt.:wt.	HPH ^2^ Pressure (psi)
MCC	None	Neat MCC	5000
EU-1:8-5k	EU	1:8	5000
TY-1:8-5k	TY	1:8	5000
TY-1:1-10k	TY	1:1	10,000
EU-1:1-10k	EU	1:1	10,000
TY-1:8-10k	TY	1:8	10,000
EU-1:8-10k	EU	1:8	10,000
TY-1:8-20k	TY	1:8	20,000
EU-1:8-20k	EU	1:8	20,000

^1^ EO—essential oil, ^2^ HPH—high pressure homogenization.

## Data Availability

Not applicable.

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
