# Peer review of "Encapsulation of Thymol and Eugenol Essential Oils Using Unmodified Cellulose: Preparation and Characterization"

_polymers, 2022, doi:10.3390/polym15010095_

Round 1

Reviewer 1 Report

1. For Figure 6 provides colorful FTIR data to make it clearer and readable.

2. Fig 7 remove the gridlines of the graph and improve the text quality of the axis.

3. Authors should calculate the antibacterial activity of the emulsion as EOs have bacteriocidal activity.

4. In the abstract and conclusion part describe which emulsion has better characteristics EU or TY?

Reviewer 2 Report

The paper deals with an emulsification process which is used for encapsulation essential oils within an un modified cellulose structure. The manuscript is adequate for this journal but it still has certain weaknesses that should be addressed before publishing. So some further explanations and revisions are necessary, considering the following points:

- Authors mentions in abstract and other section that the emulsification technique is novel....however, various methods of encapsulation already exist. Even with cellulose-based materials by applying the well-known “Pickering emulsion”.

Although authors mentions some of these aspects, the novelty of their work is not still clear compared to other published works.

They should explain in more detail why the use of unmodified cellulose..... This aspect is essential and it seems the key and one of the most important point in the manuscript. Thus it should be included in the tittle, in the abstract and so on.

-The authors affirms the as-prepared cellulose has amphiphilic character.  Can they include references with this info? 

- The encapsulation of EO within cellulose was made using ratios of 1:1 and 1:8. Why?

-  Zeta potential measuremts are highly recommended to study the electrostatic interactions between the cellulose and EO as well as the effect of pickering emulsion during the preparation of  the formulated blends.

-Additional explanations of the emulsification technique are required at the begininig of the results section.  How does the cellulose storage the EOs?

- Authors have identified the peaks of the FTIR spectrum assigning each one to a functional group. However, there is no additional information clears the incorporation of the EOs into cellulose microcapsules.... No peaks shifts, no presence/absence of new peaks. How the foundation of encapsulation process is? Physical, Chemical, other idea???

-Picture 6 should be edited. The range from 5000-4000 cm-1 should be removed. There is no info. The most relevant range is at lower wavenumber. Please, focus on this part

- Figure 7 shows the Mold development observations. How authors can explain the differences in the % mold develop (release) between the TY and EU at the same ratio? What are their physico-chemical reasons of this phenomena? Maybe molecular size? Maybe the affinity with cellulose?

Authors must include results and references of this type of assays comparing with other results in literature.

-The list of references is poor. Some of them is even old. It would be interesting review this section and complete with further cites

Reviewer 3 Report

Title: Encapsulation of Thymol and Eugenol Essential Oils using Cellulose: Preparation and Characterization

Manuscript ID: Polymers-2007347

Comments:

This study aimed at encapsulating thymol and eugenol essential oils with cellulose as the shell material to form stable aqueous emulsions using a novel encapsulating technique without additional surfactants. While the goal is interesting, the execution of the methodology is not convincing enough.

1. Methodology

The authors refer to their method as a novel encapsulating technique. However, the authors fail to elucidate the novelty of this technique clearly. The authors should provide clarity on this.

The authors only report two cellulose: EO ratios, 1:1 and 1:8. How did the authors choose these two ratios? Was there optimization done to aid the selection of only these particular ratios? The approach of just selecting two ratios without a scientific reason why contributes to a highly flawed methodology for lack of optimization. This renders some claims made by the authors, such as the 1:8 ratio being the best, questionable, and scientifically unsound. In addition, the authors wrote, "the differences with the anti-mold activity can be explained by the amount of EO in the emulsion as a result of cellulose:EO ratio emulsion with a 1:8 ratio showing improved anti-mold activity". It is obvious that increasing the amount of EO will improve the anti-mold activity. Again, the lack-of of optimization or different ratios renders this claim an obvious observation than any unique finding.

In summary, to be able to draw any significant findings and conclusions from this work, the authors need to present an optimized methodology or make a clear supplement reasoning behind this approach as presented.

2. Characterisation

The authors present a more analytical characterization of the emulsions developed. However, the authors mention that a controlled release mechanism of the encapsulated EO is important but presents only emulsion stability and anti-mold activity analysis. The authors should provide a study of the EO release rate.

Also, can the authors provide a comparison table for their findings to that of others made by different techniques? 

2. Grammar

The overall language could be improved. Just to mention a few most recurring issues that can be revised;

For sentences containing a series of three or more words, phrases, or clauses, consider inserting a comma to separate the elements. Examples are lines 15, 19, 30, 34, 36, and many others all over the entire manuscript.

Article usage problem: The authors have omitted the article ~the~ in many cases.

Incorrect prepositions such as composed by instead of composed of. 

Cases of misspellings, to mention a few, penicillum instead of penicillium, aspergilus instead of aspergillus, etc. 

Round 2

Reviewer 1 Report

Accept

Author Response

We thank you for your prompt and supportive review .

Reviewer 3 Report

Title: Encapsulation of Thymol and Eugenol Essential Oils Using Unmodified Cellulose: Preparation and Characterization 

Manuscript ID: Polymers-2007347

Comments

The authors have addressed my concerns and those of the other reviewers. The abstract now contains all the elements of an abstract, enhancing the reader's comprehension. 

The introduction has been revised with better clarity on the scope and the novelty of the work.

The authors have also tried to remove claims that are not justifiable or backed by their results.

The overall language and grammar have been revised, enhancing the manuscript's overall quality and readability.

A general recommendation is that the authors look more into experimental parameter optimization's importance. Though the authors have explained their reason for choosing a ratio of 1:1 for cellulose: EO, they haven't been able to offer an apparent, experimentally sound reason why they decided on a 1:8 ratio but noted that they aimed for a higher ratio. By experimental rationale, the optimal higher ratio could be any other than just 1:8, hence why optimization of any experiment is vital. 

Author Response

We thank you for pointing out the need to clarify the reason for choosing the cellulose:EO ratios in this study.

The relevant paragraph (in section 3: Results and discussion) was modified in this revision, as follows (as well as in "Track changes" mode in the revised manuscript):

Microcapsules of EOs encapsulated by a cellulose coating were fabricated at two cellulose:EO weight ratios, using HPH at three different pressures, as detailed in Table 1. Previous studies have shown that the main effects of the cellulose:oil ratio are controlling the size and shell thickness of the cellulose encapsulated particles. In these studies,  emulsions were fabricated by the same procedure used in the current research [30]–[32]. A thick coating was desired in the previous studies, since enzymatic cellulose hydrolysis was considered. For this aim, a 1:1 cellulose:oil ratio was found to be most suitable. In contrast, the current study sought to generate a thinner coating for the purpose of EO encapsulation and subsequent release. A higher EO loading, i.e., a higher cellulose:EO ratio was thus required, while avoiding excessive EO content, which may cause a significant amount of unencapsulated EO. Therefore, emulsions fabricated with 1:1 and 1:8 cellulose:EO ratios were chosen.